# An Exploration of Nurses’ Experience Following a Face-to-Face or Web-Based Intervention on Patient Deterioration

**DOI:** 10.3390/healthcare11243112

**Published:** 2023-12-07

**Authors:** Jeong-Ah Kim, Linda K. Jones, Daniel Terry, Cliff Connell

**Affiliations:** 1School of Nursing, Paramedicine and Healthcare Sciences, Charles Sturt University, Bathurst, NSW 2795, Australia; linjones@csu.edu.au; 2School of Nursing and Midwifery, University of Southern Queensland, Ipswich, QLD 4305, Australia; daniel.terry@unisq.edu.au; 3School of Nursing and Midwifery, Monash University, Clayton, VIC 3800, Australia; clifford.connell@monash.edu.au

**Keywords:** clinical simulation, web-based intervention, face-to-face intervention, patient deterioration, patient safety, nurses, FIRST^2^ACT

## Abstract

A web-based clinical simulation program, known as FIRST^2^ACT (Feedback Incorporating Review and Simulation Techniques to Act on Clinical Trends), was designed to increase the efficacy of clinicians’ actions in the recognition and immediate response to a patient’s deterioration. This study, which was nested in a larger mixed method project, used ten focus groups (*n* = 65) of graduate, enrolled, registered nurses, associate nurse unit managers, and general managers/educators/coordinators from four different institutions to investigate whether nurses felt their practice was influenced by participating in either a face-to-face or web-based simulation educational programme about patient deterioration. The results indicate that individuals who were less “tech-savvy” appreciated the flexibility of web-based learning, which increased their confidence. Face-to-face students appreciated self-reflection through performance evaluation. While face-to-face simulations were unable to completely duplicate symptoms, they showed nurses’ adaptability. Both interventions enhanced clinical practice by improving documentation and replies while also boosting confidence and competence. Web learners initially experienced tech-related anxiety, which gradually subsided, demonstrating healthcare professionals’ resilience to new learning approaches. Overall, the study highlighted the advantages and challenges of web-based and face-to-face education in clinical practice, emphasising the importance of adaptability and reflective learning for healthcare professionals. Further exploration of specific topics is required to improve practice, encourage knowledge sharing among colleagues, and improve early detection of patient deterioration.

## 1. Introduction

Early intervention by healthcare staff in the detection of a patient’s changing health status reduces the risk of a medical emergency [1]. It is clear that many cues indicative of a person’s deterioration are overlooked or missed altogether [2]. There can also be confusion about which clinical indicators should be recorded. 

There are widespread concerns regarding the treatment of patient deterioration, which has resulted in the formation of a body of data known as the “failure to rescue” literature [3]. It is acknowledged that nursing personnel may ignore indicators of deterioration and may be hesitant to seek help when it is required. While Medical Emergency Teams, also known as Rapid Response Teams, have shown advances in the care of critically deteriorating patients, it is critical for first responders to have the skills essential to effectively increase patient safety [3]. This is in line with the core safety and patient care concepts that form the basis of nursing standards and procedures. 

To effectively increase patient safety, it is crucial to ensure that first responders are adequately educated and equipped with the required capabilities, and that these are in line with the core safety and patient care concepts that form the basis of nursing standards and procedures [3]. To facilitate this learning, simulation is increasingly being used in nursing as a teaching and learning strategy to develop nurses’ knowledge and skills [4]. This teaching and learning strategy encompass activities that resemble circumstances that would be experienced during work-integrated learning that enables the demonstration of clinical actions and decision-making [5]. The evaluation of this strategy has shown improved self-efficacy, strong practitioner satisfaction, and improved confidence and/or critical thinking [3,5].

To support compliance with Standard 9 (National Safety and Quality in Healthcare Framework) [1], which requires competency in managing the deteriorating patient, a web-based clinical simulation program described in a theory-based model by Buykx et al. [6], known as Feedback Incorporating Review and Simulation Techniques to Act on Clinical Trends (FIRST^2^ACT), was developed. FIRST^2^ACT is a clinical simulation program that is offered either face-to-face or as a web-based program. The program is designed to increase the efficacy of clinicians’ actions in the recognition and immediate response to a patient’s deterioration. Face-to-face and web-based versions have demonstrated impact on educational outcomes and clinical performance with regard to increasing participants’ knowledge and prompt call to action [2]. Using the web-based or e-learning approaches ensured important information was delivered [7,8], reflecting desired outcomes [9] and reducing the difficulties associated with creating time for continuing professional development.

In 2016, a mixed method study (this study was registered as clinical trial: see https://www.anzctr.org.au/Trial/Registration/TrialReview.aspx?id=370425, ANZCTR Trial registration: ACTRN12616000468426, 8 April 2016) was designed to demonstrate the impact of clinical simulation on improved early detection of patients’ deteriorating health status. This larger research project was designed to compare the effectiveness of two forms of simulation education, face-to-face (F2F) versus web-based (WB), facilitating nurses’ ability to detect and manage patient deterioration. During the FIRST^2^ACT face-to-face simulation, assessment tests were conducted, simulation occurred, and feedback techniques were delivered to individual participants by a team of facilitators over a one-and-a-half to two-hour period. The web-based FIRST^2^ACT program constituted an online learning package. All participants with different skill mixes completed three contrasting, eight-minute simulation exercises that included patient deterioration at the midpoint. Acute myocardial infarction, hypovolaemia, and chronic obstructive pulmonary disease comprised the patient scenarios. In the face-to-face experience, video recordings of participants’ actions occurred. This initiative referred to as “photo elicitation” [10,11] provided an audio-visual record used as a reflective account of participants’ decision-making. Individual feedback was given by an instructor. 

The FIRST^2^ACT web-based educational package included a series of three professionally video-recorded scenarios using specialist actors as patients. A “mouse over” function enabled participants to click on an action, for example, lay the patient flat, administer oxygen, or take an ECG. On completion, results were provided with automated feedback on performance outcomes and, where the pass mark was reached, a certificate was issued. Participants could make as many attempts as required to achieve mastery. 

This study was undertaken in four different hospital contexts, three rural and one metropolitan, with nurses working primarily in acute care. Nurses participated in either web-based (two sites) or face-to-face (two sites) learning. The mixed methods study aimed to establish which educational approach best served to enhance practitioners’ skill development in the early detection and management of patient deterioration and provide participants with opportunities to experience alternative pedagogies building on individual knowledge and experience [12]. As part of the overall mixed method study, clinical evidence gathered from patient records, pre-questionnaires, and post-intervention evaluation rating scales helped to demonstrate that the two educational interventions used did have an impact on practice [12]. 

What was omitted from this aspect of the mixed methods study, however, was the nurses’ experience of either program. Thus, what is highlighted in this paper is the nested qualitative dimension of the larger mixed methods project. In this nested component, focus groups were conducted to capture participants’ experiential insights in undertaking either the web-based program or the face-to-face version of the educational interventions. The findings gleaned from this qualitative aspect of the overall study will be used to inform future changes to the FIRST^2^ACT program. To better understand participants’ experience of the two different educational interventions, focus groups were held in the institutions in which the interventions took place. Within this context, the aim of this paper, therefore, is to explore the insights and experiences of nurses who had undertaken the FIRST^2^ACT educational intervention.

## 2. Materials and Methods

Following ethical approval from all institutions involved, qualitative data were collected using focus groups aiming to uncover participants’ experiences and the perceived influence the intervention had on their clinical practice. A purposive sample of participants who had completed either of the educational interventions was invited to participate in one of ten facilitated focus groups. The data were collected three months post-intervention (September–October 2016). Participants were provided with Participant Information and Consent Forms to complete and sign prior to the commencement of each focus group held at their institution. At the beginning of each group, participants were reminded they had the option to withdraw their participation up until the point of starting the focus group. Beyond that, it would be difficult to specifically identify an individual’s data from the audio recording and transcript to remove them. Participants were also provided with the opportunity to stop and restart the audio recordings at any time. To maintain the integrity of the study, six of the ten focus groups conducted involved participants who had undertaken the face-to-face intervention (31 participants), and the remaining four focus groups (34 participants) were held with those completing the web-based program. This allocation strategy ensures that there is no cross-contamination between the groups. 

The participating hospitals in regional and metropolitan Victoria range in size, where the larger hospitals employ approximately 160 nurses, and the remaining two smaller hospitals employ approximately 85 nurses each. In total, 65 of the identified nurses that required training volunteered to participate in the focus groups (82% participation rate). Participants comprised graduate nurses (*n* = 7) (1st-year post-registration), enrolled nurses (*n* = 17), registered nurses (*n* = 31), associate nurse unit managers (ANUM) (*n* = 6), and general managers/educator/learning program coordinators (*n* = 4). In most focus groups, the skill mix varied, while ANUMs were present in a third of all focus groups, which may have had some impact on the overall dialogue.

All participants were female and ranged in age from 21 to 62 years of age. Ninety percent spoke English as their first language. Among their qualifications, 36% were hospital training, 56% obtained a bachelor’s degree in nursing, while 8% had undertaken post-graduate training. The clinical practise experience, which was largely in the acute medical sector, ranged from 1 to 43 years. The wide representation of team members offers a thorough view of the issue under consideration. Different roles and degrees of expertise in the healthcare settings provided a broad perspective on the impact and experience of educational interventions on clinical practice and patient care.

Each focus group lasted approximately 60 min and was audio-recorded and transcribed verbatim. The research team designed the focus group technique to be adaptable and flexible rather than pre-structuring a defined list of questions, The questions were formed naturally during the interviews, led by the topic’s themes and the participants’ reactions [13]. This strategy enabled a more in-depth study of the participants’ experiences and viewpoints, resulting in rich and contextually relevant discussions. Consensus was not the aim, as the focus group discussion sought to provide diverse opinions, which was encouraged and is included in the analysis. All the focus group sessions were conducted by two researchers in a single round, and both had prior experience leading focus groups. 

Questions used for this research were developed from the literature. Content validity was ensured through an expert panel which encompassed the researchers and key stakeholders from industry. The interview schedule included starter questions about the following themes:Form of program completed (either face-to-face or web-based).Fidelity of simulation.Clinical applicability of the program they completed.Educational outcomes in terms of changes to practice.Possible improvements to the program.Reflections on their practice pre- and post-intervention.

Field notes (as part of the reflective practice) were also recorded in two researchers’ personal journals and formed part of the data. To support rigour, transparency of documentation, audit trails, and authenticity of commentary, the researchers’ reflections in asking participants the questions and then inviting commentary on colleagues’ positions to ensure clarity in meaning were included.

Focus group transcripts and field notes recorded in journals by the two researchers were independently analysed. Data were analysed using coding based on language and meaning and from which tacit and embodied understandings informed the subthemes and themes. Field notes provided additional information about participants’ engagement, non-verbal details, tracking ideas, and reflexive questions to pose when the data were analysed, including questions about power relationships where mixed-skill focus groups were taken for granted. 

A coding format was initially used to locate and cluster verbal texts associated with the questions. Texts were then further examined and re-clustered to expose underlying or alternative ideas. As additional clustering occurred, new understandings emerged in the language as it was exposed by peeling back or unfolding different meanings, in which the participants’ language served to implicate aspects of practice [14]. The interpretive lens facilitated the initial examination of textual experiences that were then reconstructed to enable the development of subjectivities revealing participants’ tacit and embodied understandings of their practice interventions [13]. These understandings were then grouped into the sub and then common or core themes reflecting multiple dimensions of meaning whilst articulating tacit knowledge and taken-for-granted assumptions in practice [15]. This interpretative lens helped to highlight the implicit and useful knowledge that practitioners have and use in their work [14]. Meanings were also “checked” with subsequent focus groups to increase the depth of understanding, establishing credibility, resonance, and authenticity [13]. 

For the purpose of clarity, coded details at the end of each quote make clear the program involved, where web is used for participants who undertook the web-based program, while for participants who engaged in the face-to-face version, the code f2f is used. In addition, to maintain confidentiality and for the ease of identifiability of participants by the role they were undertaking within the health service, these identifiable characteristics are not disclosed. 

The trustworthiness of the research methods also requires reflexivity [16]. The significance that reflexivity plays in the rigor and validity of qualitative research and the recognition of researchers by reflexive research are embedded within the setting, context, and culture they are seeking to understand and analyse—they are an instrument of the research. In addition, reflexivity is a process where the researcher needs to self-assess and recognise their own subjectivity, preconceptions, motivation, and theoretical foundations within the research process [16]. Within this context, it is vital to ensure all researchers are critically reflective of how they self-locate their position and interest throughout the research process. 

For transparency, all researchers were nurses, some with emergency, occupation, and workforce development backgrounds. However, all possessed a shared vision that clinical simulation has a positive impact on clinical learning and professional development. As such, each researcher was to ensure that the knowledge production and the outcomes of the research were not biased, which would lead to fewer distortions of the research accounts due to personal backgrounds, perceptions, and interests. As part of this process, each researcher was to review and analyse the data in such a way that they avoided any previous notion or anticipation of findings and eliminate portraying personal opinions as the findings emerged. 

Lastly, it must be noted that the paper has been through a lengthy peer review process since data were collected, which encompassed several rounds of input and revision due to the complexity of this research and the possible consequences of the findings.

## 3. Results

Among the 65 participants, three core themes emerged from the data, which encapsulated (1) the structure of the program and its impact on practice, (2) surveillance and patient deterioration, and (3) the tacit knowledge informing clinical judgement. Each of these themes and sub-themes is discussed in detail.

### 3.1. Structure of the Program and Impact on Practice

One of the questions faced in the overall study was to try and gauge whether web-based education was more effective in learning about clinical practice than face-to-face interventions. As part of this quest, the focus group participants were invited to explore what they felt about the structure of the program and whether it had benefitted their practice. Participants from both programs highlighted certain aspects of the experience, including (a) the benefits of ongoing learning; (b) reflection on the process; (c) mirroring the everyday world; (d) how simulation affected the participant’s clinical practice with subsequent impact on patient care; (e) anxieties experienced undertaking the program.

#### 3.1.1. Benefits of Ongoing Learning 

Comments relating to the web-based program varied. There were participants who enjoyed the opportunities to engage with the program because *“a lot of us are a little bit older, less techno-savvy, and perhaps learn face-to-face better than—”* (wb). Given that the dominant age range for registered and enrolled nurses practising is 50–59 years (24.7% of the nursing workforce) [17], it is not surprising that many nurses felt they were not as computer-literate as younger colleagues. Nevertheless, the online program was considered *“…good…I like that business. Because you can do it at your own pace”* (wb). Moreover, the opportunity to achieve mastery with self-pacing *“increases the confidence”* (wb). This indicates that there were advantages and challenges associated with both education modes.

With participants in the face-to-face program, because of the generic nature of the training in terms of how it might be used in multiple healthcare environments, there was a noticeable sense of discomfort in terms of the scenarios reflecting particular contexts of practice which did not apply to all. The most common was the public/private hospital divide and how these two distinctly different organizations engage in caring practices that affect the deteriorating patient. This is illustrated in the following quotes.


*It’s a few fundamentals that I think [are] quite different between private hospitals and public hospitals; so how we work wasn’t really differentiated. Yes, the scenarios would all be the same, but the way we work within the hospital was very different…*
(f2f)


*I think it’s more perhaps suited for a public hospital than a private. Like I work public and private, and it was very—probably more like what would happen in a public hospital, having the doctor right there and [click] on standby. Whereas here [in private] you don’t.*
(f2f)

Not having a medical practitioner on-site twenty-four hours a day in private hospitals requires the registered nurse to use clinical understandings that are often derived from years of experience, expertise [18], and tacit knowledge [19,20,21,22]. This experience, expertise, and knowledge is recognised and supported by programs such as this one. *“I’ve been nursing for a hell of a long time, so it was more reinforcing”* (f2f). Indeed, professional development in healthcare institutions could benefit by using either format. In other words,


*What you should do when you should see that deterioration, just a refresher that sometimes you might not notice something that you should.*
(wb)

#### 3.1.2. Reflections on the Process

Looking back and reflecting on their experiences, participants in the face-to-face program saw its value for a number of reasons: 


*It was a positive experience. It was a powerful learning experience. So, to be able to self-reflect afterwards and yeah, talk about what we could have done and what we didn’t do and what we did do.*
(f2f)

In addition, the filming of the participant’s performance enabled the individual to stand back and view their engagement in the simulated situation. This performativity of the nurse, while not real, did approximate reality. 


*It’s different when you’re in the situation, you just go and do things. Whereas being outside and then looking at the film was another experience that you could pick up on things that in the moment—that you weren’t necessarily properly concentrating on or thinking of. So, it gave you another perspective being outside, looking in and not part of the situation.*
(f2f)

Furthermore, examining how one acted generated confidence and provided ways to think about how one acted in different situations using experiential understandings. *“It also builds your confidence to know that you actually have got that knowledge…you are doing the right thing when you get your feedback”* (wb). These acts of surveillance highlight the practitioner’s self-criticism or, perhaps more importantly, the absence of critique from an instructor. *“I remember at the time feeling frustrated that there wasn’t more feedback”* (f2f). In such cases, it is possible to suggest that nurses have become enculturated into understanding that external sources of approval carry greater weight than self-recognition of one’s accomplishments. Unquestioned understandings appear as common sense, an acknowledgement of “this is what is, and this is what needs to happen”. In these circumstances, there is no external voice, rather, one is looking at the self and gauging one’s performance on what one understands the protocol to be. This shift in thinking is significant in terms of self-engagement and reflection and has an impact on one’s practice standards. Nevertheless, the importance of commentary from an authority (the instructor) adds weight to the disciplinary voice [23] of oneself to ensure the right action occurs. 


*I think that would be the most valuable aspect of the whole training—the scenarios were excellent—and made us think quickly which is really important. It’s ultimately having a little bit more feedback surrounding our actions and areas to improve.*
(f2f)

Thinking in correctional terms, “wrong/right”, is not helpful. What it really overlooks is that practice is not a question of binary distinctions—good and bad. Rather, it is a complex tentative process of making decisions based on the fluidity of data where the practitioner makes informed judgements relative to their experience and practice understandings. 

#### 3.1.3. Mirroring the Everyday World

Participants raised concerns about the limitations of the simulation. In particular, when face-to-face with an actor, they had to ask the actor what they were feeling and other questions about the deteriorating experience. As the actors were unable to mirror changes in their vital signs, some of the constraints of the learning were apparent. *“When [the patient] is an actor—you have to actually ask are they sweating, are they clammy. That makes it obviously not real to me anyway”* (f2f). In this regard, predictive or anticipatory knowledge helps to show the individual what they are expecting, and this, too, depicts the nature of the nurse’s knowledge or, indeed, pattern recognition. The importance of mirroring everyday reality is problematic when the actor playing the doctor is unable to portray the material world. 


*Because you knew we were hands on and we were all straight in there and we just found it weird that this doctor said I don’t know, I don’t know, you know what the answer was, but can we tell you what it is…*
(f2f)

While this was regarded as frustrating, it did emphasize the depth of the nurse’s knowledge in the care provision of the deteriorating patient. Clear demarcation lines between the modus operandi may be apparent in some clinical contexts but will vary in others.

The knowledge of processes in the everyday reality of practice was not well-mirrored in the web-based experience. In part, this may have been due to what might well be understood as a one-dimensional experience that is predicated on “right” choices and singular activity.


*It was frustrating because if you’re in a clinical situation you can do two things at once. While you’re, say, taking an ECG you’d be asking the patient about their family history or anything like that. You couldn’t do that. You had to specifically do one thing and then you’re locked out for that length of time, which seemed, when the clock’s ticking, it seemed like a long time. So, I found it very frustrating.*
(wb)

Sequential events portrayed as singular moments that together provide a picture of the deteriorating patient help to reduce the complexity of the situation. Recognition of patterns have consequences in early detection as a logical flow of outcomes given the nurses’ responses. At the same time, slowing down these events assists health professionals to more closely watch these patterns and make sense of the meanings in terms of clinical importance and, thus, impact. 

#### 3.1.4. How Simulation Affected Their Clinical Practice

The extent to which the program made a difference to participants’ practice was emphasised in all focus groups. Improved documentation that prompted recognition at strategic moments would “trigger” a call for help. *“Just document, document everything”* (f2f). Keeping the records up to date and ensuring that vital signs were being recorded enhanced clinical impact. In addition,


*I think it also helped identify the gaps on the observations sheets and what we were and weren’t doing correctly. I think I’ve even seen now back on the ward that people are utilising them…the obs charts now…and documenting it properly.*
(f2f)

The web-based program also enabled repetition, building one’s repertoire of skills and enhancing confidence. 

#### 3.1.5. Anxieties Experienced Undertaking the Program

Many participants spoke about their anxiety in undertaking the web-based program, in large part because of their lack of familiarity with computer technologies. They commented on literacy, having to establish how the program worked, the absence of collegial support, and, thus, the risk of making a mistake. Those who undertook the web-based program found their anxiety levels were ameliorated with increasing knowledge of the format and opportunities to practice. This was particularly important to nurses who had limited computer skills or minimal exposure to using one. *“People who weren’t computer literate were more anxious”* (wb) and because they undertook the program on their own, in the absence of colleagues, *“…you were doing everything on your own. You didn’t have that support you felt”* (wb). Perhaps the presence of colleagues to facilitate the experience and affirm decisions in the “real situation” works to build self-assurance, reducing the levels of anxiety.

Nurses who undertook the web-based program found some anxious moments in completing it in a timely manner, or, at least, each of the various stages. Having to learn how the program worked did increase some nurses’ anxiety because they were worried about providing the responses in the “correct” way. Nevertheless, being able to address the issues as they arose “*helped the problem-solving process, yeah. I think it facilitated that and sort of helped guide you through what you’d sort of to-do mentally anyway”* (f2f).

### 3.2. Surveillance and Patient Deterioration

Organisations have their own systems of surveillance for ensuring that practitioners comply with standards that form aspects of the discipline’s practice standards, as well as those that are institutionally driven but frequently taken for granted by staff. Organising systems, therefore, include not only many reified practices but also overt protocols. The complexity of managing the deteriorating patient is best understood by participants in *“[clear expectations, clear guidelines and expectations from the medical team would absolutely go a long way I think if there’s the clear direction it eliminates all of that uncertainty…]”* (f2f). The context of this comment was one in which the participant found themselves “directing” the activities and providing advice to the medical officer. Many nurses in the study had a practice background in intensive care and/or had undertaken a critical care course (field notes). Well-informed insights gained in the past provided the ongoing development of practice wisdom. 

The clarity in communicating the nature of the patient’s status is vital. “A good handover is key. So, it’s making sure that that continuity in communicating is always open and that information is passed on [about a patient]” (f2f). It could well be that the nuances in a patient’s clinical status are not recognised by staff. As one accumulates these understandings, one’s intuitive knowledge develops over time. The less experienced would not perceptually recognise the small subtle aesthetic [24,25] shifts in status. 


*There are quite a few altered conscious states that do or don’t get called, it depends on different situations. People that are in palliative care situations for instance still have MET calls that…hasn’t been altered yet. They’re in that phase that we wouldn’t call MET call.*
(wb)

The documentary requirements noted by staff facilitated the initiation of a MET (Medical Emergency Team) call. There were instances when the patient appeared to be “at risk” of having reached the MET criteria. In situations where the patient was receiving palliation, however, no MET call occurred despite the lack of clear documentation or shared information. *“Someone’s deteriorating and there’s been a decision that they won’t take any further action, but it hasn’t been clearly documented.”* (wb) One wonders whether the nurses’ local knowledge informed their practice decisions based on aesthetic understandings (for example [26,27]) and the various interpretive positions they may have taken. For nurses newer to the field, this intuitive and maybe relational understanding between colleagues could be constructed as a failure to communicate. 

Participants in one focus group spoke about some of the difficulties they faced in noticing that the MET criteria required the initiation of action. In these instances, nurses tended to seek advice from their supervisory team, not relying on their own clinical judgement. Nurses’ actions varied in response to vital signs’ chart criteria for triggering assistance. *“If you run it past your supervisor or whoever’s in charge and they say no—[there is] nothing you can do about it”* (wb). On occasion, however, *“[you might get the [Critical Care Unit] (CCU) liaison to come around and then you’ve got your buddy helper and you might manage it before it gets to be a MET call]”* (f2f). Surveillance appeared to be multifaceted with clear discriminatory guidelines informing nurses of when to act, thus reducing risk. In some respects, however, it deskilled nurses by not fully calling on their scope of practice. There are, therefore, tensions in managing risk that potentially erodes the nurses’ knowledge and practice understandings. As this nurse who had engaged in the face-to-face program commented, *“we were doing everything that needed to be done and we transferred the patient out. We didn’t need to do all the bells and whistles”* (f2f). This example highlights the knowledge that more experienced nurses use to support the safety of the patient prior to generating a MET call. 

### 3.3. Tacit Knowledge in Clinical Judgement

The “Between the Flags” system, introduced in 2010 to improve the identification and response to deteriorating patients, is widely acknowledged in New South Wales, Australia [28]. This system involves strict adherence to vital signs charting protocols, triggering a MET call when specific criteria are met. However, healthcare professionals recognise that such strict adherence may have the unintended consequence of limiting critical thinking and diminishing the flexibility in making clinical decisions [29]. Over time, this approach could potentially lead to deskilling, where healthcare providers rely solely on chart data, resulting in the decision-making process becoming automated and taken for granted. As one healthcare worker aptly expressed, *“I reckon the nurse’s clinical call often is the right call. Rather than just relying on the raw numbers”* (wb).

The result is that practice knowledge diminishes, and the responsibility for the patient gets allocated to someone in a more senior position or a nurse who has undertaken additional education. *“It’s about the management which—that reiterating that we’re managing and what we need to do and what’s right rather than we’ve called the MET, it’s [now] someone else’s responsibility”* (wb). Nonetheless, the charts create an opportunity to learn the patterns of deterioration and provide nurses with a valuable resource. 


*We’re not always thinking of why it is going down; we’re thinking oh, it’s hit the yellow. I just need to call. We’re not thinking well, it’s going down…I think they might be a bit dry. We need to think of what interventions we need to do at that stage…*
(wb)

Learning these patterns develops one’s intuition. This is a form of tacit knowledge derived from experiential meaning [30] that, when called upon, shifts one’s thinking from a task-based observation of noting changes on the chart, to making meaning of the temporal situation.


*If you’re concerned about a patient. I mean regardless, I would say, for me, it would be gut instinct. If I’m concerned and the obs look okay—sometimes the observations aren’t really going to be your prompt to call.*
(f2f)

Well-constructed feedback from the authorial voice of the institution or taken-for-granted assumptions directing self-surveillance impact one’s confidence. Taking a different position could result in retribution. *“Sometimes we might be discouraged from making that right decision when we should because we’ve been bitten a couple of times where we’ve rung [or called] them [doctors]”* (wb). According to several participants who work in the private sector, the ongoing experience of antagonistic relationships between nurses and doctors continues. 

## 4. Discussion

Connecting the purposes with the perspectives from which the study originated and the cultural environments in which this took place adds to the veracity of the research by reportedly making a difference in participants’ practice [31]. Together, resonances of information, meanings generated by participants, researchers, and the reader help to establish trustworthiness. Collected successive dialogue of a similar nature highlights the impact of the two different approaches to learning and patient outcomes. Comments such as “*I’m definitely thinking more about the MET call situation now. If there isn’t something done already that I can do, I do it rather than just standing there and waiting…”* (f2f); *and* “*Maybe we could do both though because I thought it was good to do it individually because it made you think”* (wb) were common. Statements such as this imply the importance of providing different approaches to learning to take into account the social, relational, and individual perspectives in meaning-making, which is crucial. 

As highlighted, three core themes emerged from the data. The structure of the program and its impact on practice was found to be important in ongoing personal-professional development. In particular, how one generates new meanings from engaging the self differently was enlightening if not confronting for participants. Most participants had previous learning experiences that stood at odds with the simulation processes they encountered. One example was in the opportunities to see the self perform and then evaluate their knowledge. Visualising the moment-to-moment decision-making in the web-based program provided a forum for developing self-paced knowledge. For example, a significant implication of the program is an increased awareness of the importance of documentation, particularly when dealing with emergency situations. Yet, while documentation was regarded as a critical component to reviewing a patient’s status, it did not necessarily inform changes to practice where experiential knowledge called for alternatives such as “wait and see”, or where the context was part of the patient’s normal pattern. These situations require further exploration. 

Reflecting on the process of simulation and gauging action helped to reframe problem-solving and the decisions taken by individual nurses in the web-based scenarios. Initially, anxious moments were in the timing of decisions and making sure one acted appropriately in completing the program. Here, self-surveillance highlighted taken-for-granted assumptions that were hitherto unobserved. Tacit knowledge is developed from enhanced skill performance and augmented clinical knowledge [32]. Moreover, the web-based program enabled participants to have an opportunity to repeat the simulation, increasing self-confidence, but it had less impact in terms of team knowledge and working together when a MET call was initiated. The focus of each session was on individual attainment rather than developing shared knowledge. While this approach assists the practitioner to increase their knowledge, there remain questions about ownership of that information when it is privatised.

The notion of tacit knowledge informing clinical decision-making remains problematic. Tacit knowledge, for instance, reflects the personal understandings nurses use to inform their practice. Traditional meanings stem from, for example, Carper’s [26] work on fundamental patterns of knowing in nursing. Carper argued (as have others subsequently, such as Benner [10], Benner and Wrubel [21]) that tacit knowledge is a component of aesthetic understanding and expertise [18], so that meaning-making derives from the merging of sequences into a picture to make sense of what is happening. The struggle to acquire this knowledge takes time. Yang and Thompson [33], however, oppose this view and point out that nurses’ judgement might not improve with increasing years in practice. In another study, Cioffi [34] reports that nurses considered their ability to recognise patients’ health status to be heavily grounded in past experiences. One wonders if the notions of performativity are contained in nursing knowledge or just in skilled action. Notions of performativity should be foundational in intuitive understandings embedded in nursing knowledge as a precondition for the right action. Linder and Pulsipher [35] claim that simulation “training” assists the development of critical thinking and clinical reasoning, not only clinical judgement. Their findings are not completely parallel to this study because of nurses’ perception of maximising clinical judgement which some participants saw as being systematically reduced when the usage of the standardised observation charts dictates what nurses need to do and when. 

This study focussed primarily on tacit procedural knowledge development [36]. Taking Held’s [37] interpretation of Habermas’ [38] thesis on knowledge and human interests, educators need to enhance strategies that support grappling with the deeper meanings associated with forms of rational action—those aspects of performance that are dialectically related to language, culture, and social relations [37]. In this way, there is an avoidance of a structural-functionalist position in which cause–effect relationships are exemplified, rendering invisible opportunities for understanding [39] as well as accessing that knowledge that extends beyond singular scientific claims to truth or technical rationality [40]. Schatzki [41], for example, argues that practice occurs in a context in which things or entities are situated within a set of special relationships. These arrangements of elements convey meaning. For nurses, as for other health practitioners, particular relationships between things depict patterns suggestive of an interpretive identity. Meanings are, therefore, reflective of what the individual understands this identity to be within a set of social arrangements or relationships. Moreover, it is the ability of the individual to work out what makes sense in particular circumstances that makes their performance intelligible [41]. Especially pertinent if nurses are being encouraged to change their practice, the notion of action and sense-making is the consideration of the socio-relational, cultural, material, and historic elements in which arrangements of action occur [14]. Deeper meanings contained within these practice sites provide a platform for skill acquisition situated in complex practices of purposeful action. If a change is required, however, and individual practice is transformed, then the social world in which one’s engagement occurs also needs attention as the action takes place within social circumstances [42,43] in a material world in which one is shaped by, and shapes, the pervasiveness of pre-existing discourses. Therefore, embodied engagement involves learning across a material world involving cultural and relational ways of being. While there could be a tendency to focus on skill attainment alone, there is a risk of omitting taken-for-granted discourses where deeper meanings associated with rational action or sense-making might be realised.

Participants found that observation charts discouraged their ability to think critically, a situation they felt stemmed from the assumption that health professionals with more clinical experience and knowledge have better clinical judgement. Examples such as informing the doctor about a patient’s condition were deemed the most appropriate course of action for those in the face-to-face program. The charts requiring nurses to take action at particular points in the patient’s trajectory serve to reduce institutional risk. This study found that a combination of approaches to facilitate the opportunities to further develop practice understandings that on one hand address questions about performance, and on the other, engage participants in sharing knowledge and meaning making, is one of the ways forward. 

This study endorses the findings of Chung et al. [12] which suggest that opportunities for blended learning will facilitate ongoing education in clinical practice. However, as Kim and Lee [44] describe, cognate, affective, and psychomotor areas plus the value of simulation to the learner and the learner’s developing competency are critical to educational strategies designed to increase clinical competence. A focus on the technical only, however, will distort potential patterns of action because it fails to take into account a fuller picture of the patient’s health status. Central importance needs to be given to the cultural and relational elements in which nurses’ performativity (not just the notion of competence) can significantly affect a patient’s deterioration. Nurses’ theories in use are shaped by, and give rise to, identities in which patterns are formed, and where understandings reflect the fluidity of unstable environments in which nurses work and make sense of their realities. 

These findings would suggest that by unpacking the practice worlds of nurses to reveal assumptions, clinical judgements, cultural knowledge, and relational positions, a better understanding of situations in which nurses find themselves making decisions about which courses of action to take when a patient deteriorates will reduce anxiety and diminish self-surveillance. A further analysis of participants’ commentary, their skill level, gender, and age may also provide much-needed detail on differences in educational approaches to program development. In addition, processes of achieving consent with a staff of mixed status within the institutional hierarchy may add to closer scrutiny of the ways in which discursive patterns in the dialogue reflect power relationships in terms of the nature and substance of contributions. Put another way, power relationships underpinning the dialogue in this study may have had more influence than can be documented here. It would be interesting to separate the focus groups into staff categories such as RN or administrator to see if there was a variance in the dialogue. 

Lastly, translational work to other fields would add to a growing understanding of alternative pedagogies that take into account tacit and embodied knowledge and their implications for practitioners. In addition, this knowledge may add new light to the meaning of competency. The nurses in this study felt that both the face-to-face and the web-based programs were valuable, a situation that parallel’s Chung et al.’s [12] findings. As an ongoing educational program, both the face-to-face and web-based activities had their beneficial dimensions, adding value to nurses’ knowledge and intuitive grasp of what is happening when a patient deteriorates. This built confidence and affirmed participants’ understanding. 

### Limitations

Limitations of the study include the fact that reality was distorted by role-play; participants who were technologically challenged had to learn how the program ran before they could operate the package. Further information on the experiential knowledge of participants is also warranted to evaluate the success of the health team who have different skill sets. 

## 5. Conclusions

Nested in the larger mixed method project, this study used ten focus groups located in four separate institutions to explore whether nurse participants felt their practice was influenced by participating in either a face-to-face simulation educational program on patient deterioration or a web-based intervention. Using different pedagogical strategies draws attention to the learner’s literacy in the design of the program, a situation that requires careful consideration when working with people who have diverse skills and knowledge. Further research needs to explore how best to garner participants’ experiences to maximize meaning-making in learning where nuanced changes in a patient’s condition foreshadow deterioration. Revisiting Benner, Tanner and Chesla’s [40] approach to sharing experience via storytelling (narrative) may hold additional promise in coming to grips with the complexity of patients’ situatedness.

Attention to mirroring reality in both the web-based and face-to-face programs is vital because educational interventions will then have the capacity to bring together those arrangements and identities that comprise situational contexts, relational, political, and economic aspects of reality that have a bearing on the performativity of the practitioner, enabling them to make sense of the contexts in which they find themselves and then to act intelligibly. Further unpacking of these various texts will add to ways in which practice can be improved and greater appreciation of embodied knowledge, and tacit understandings in this context can then be shared with colleagues, promoting early recognition of patient deterioration.

## Data Availability

The data that support the findings of this study are not openly available due to reasons of sensitivity but are available from the corresponding author upon reasonable request.

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
