# Peer review of "An Exploration of Nurses’ Experience Following a Face-to-Face or Web-Based Intervention on Patient Deterioration"

_healthcare, 2023, doi:10.3390/healthcare11243112_

Round 1

Reviewer 1 Report

Comments and Suggestions for Authors

Thank you for the opportunity to review this paper.

The paper focuses on exploring nurses' experience following a face-to-face or web-based intervention on patient deterioration. This is the qualitative part of a mixed method study.

The paper is well-written with a good introduction, well explained steps in the methods of data collection, and also steps indicating rigour, transparency, and comprehensive presentation on the methodology and the paper as a whole. The use of a variety of techniques in gathering data reinforced the validity of the themes that were created through the analysis. The authors also include plenty of direct quotes to elaborate on their analysis as well as support their assertions of the statements they make in their analysis. This adds to the authenticity of the content/knowledge gathered in the study.

The paper is very systematic in the flow of information making it possible for the reader to understand the topic (that otherwise would be complex).

My two concerns are:

1). On page 5, line 199 - What is the relevance of this quote that focuses on fire and safety when the paper in on patient deterioration? Please expound on why it is included here.

2). It is paramount for researchers to do 'Reflexivity" for qualitative studies. The is means a clear explanation of the researcher(s) role as relates to their relationship with the participants, context or the topic at hand. This was not explained at all in the study. This is usually a requirement for qualitative studies given the subjective nature of strategies and methods used in gathering data as well as the data itself. Could the authors help us understand who they are in relation to this context, topic etc.?

Otherwise the findings add value to the discipline of nursing and may be a springboard for further studies as indicated in their discussion and conclusions.

Author Response

Dear Editor of Healthcare,

Title: An exploration of nurses’ experience following a face-to-face or web-based intervention on patient deterioration

Thank you for the opportunity to respond to the reviewer's comments to help strengthen this article. To ensure clarity and transparency in our response, we have prepared a table summarising the reviewers' comments and our corresponding responses. This table is included in this letter.

We believe that these revisions have significantly improved the quality and rigour of our manuscript.

We would like to thank the reviewers and the editorial team for their constructive feedback and the opportunity to further refine our research for potential publication in Healthcare.

Please feel free to contact us if you require any additional information or clarification regarding our responses or the revised manuscript.

Yours Sincerely,

Jeong-ah Kim

Reviewer 2 Report

Comments and Suggestions for Authors

Lines 494 to 505 - as they refer to future research, I recommend inserting them in the Conclusion.

Likewise, lines 511 to 515, which deal with the limitations of the study, should be included in the Conclusion

Author Response

(The authors gave the same response as above.)

Reviewer 3 Report

Comments and Suggestions for Authors

The present study has raised some concerns regarding the utilized methodology and analysis. The lack of detail in the study design and analysis poses a challenge in replicating the findings in a different context. To mitigate this, it would be beneficial to group the methodology, including the sample, methods, and type of data analysis to understand the study's underlying assumptions and limitations better. It is especially crucial to provide adequate justification for the sample size. How was consensus achieved, and how many rounds were conducted? Furthermore, the study's data analysis process lacks transparency. Can you provide more information on the development and validation of the interview guide? In focus group analysis, identifying and supporting themes is critical.

Author Response

(The authors gave the same response as above.)

Reviewer 4 Report

Comments and Suggestions for Authors

Thank you for the opportunity to review this article. This article investigated "whether nurses felt their practice was influenced by participating in either a face-to-face or web-based simulation educational programme about patient deterioration". Simulation is gaining popularity as a legitimate method of teaching in healthcare, and as a result, this is an important study. However, this article had many challenges, and these will be highlighted below:

1. Background: The background is superficial and does little to demonstrate existing knowledge about the use of simulation in healthcare, challenges and strengths. This section started by referencing one 'standard 9', but it was unclear whether this standard is universal or Australia-specific or specific to the nursing profession. Although a significant amount of time was devoted to the FIRST2ACT simulation programme, it was unclear what situation created the need for the program in the first place, why simulation was thought of as the best approach to meet the need, whether nurses alone were the intended target for the programme, whether they were consulted in the development of the program and so.

Aim: The aim states "to explore the experience of those staff members who participated in the focus groups designed to gain information on what participants felt the impact of these educational interventions had on their practice. This aim is unclear, loose and unclear how it shapes the study. For example, why use 'those staff members' if nurses were the intended participants? If the impact of stimulation was the goal, is a focus group the best method to assess impact?

Study setting: More information is needed about the study setting. Where in Australia was the study conducted? The ethical clearance numbers need to be provided. If participants "could terminate their involvement at any time during the interviews", what happened to the responses they had already given in the group setting? Considering that the participants had varying scopes of practice, what was the rationale behind mixing them in the groups?

Data collection: Did both researchers who conducted the interviews attend all focus groups? Could you provide a sample of the questions in the groups?

Findings: Could you provide the demographics of the participants? The lack of it makes it challenging to make sense of the findings. For example, statements like this, "Given the dominant age range for registered and enrolled nurses were practising is 50-59 years" do not make sense in the absence of the demographics. There is limited sign of in-depth thematic analysis and limited variations/comparison of experiences. In many cases, it feels as if the f2f groups were asked different questions from those in the web group.  This makes one wonder whether the rationale for interviewing web-based separately from f2f. For instance, why web-based said the program was good, the f2f was discussing practicalities between working in private and public hospitals. The analysis needs to be re-visited due to themes and sub-themes not always capturing the name of the theme,

Other comments: How did participants being watched during simulation impact how they engaged in simulation, and how was this addressed? Presenting themes and using literature makes the work really hard to follow, especially when there is another exhaustive discussion section.

Comments on the Quality of English Language

The manuscript needs to be professionally edited due to complex and very long sentences that obfuscate meaning.

Author Response

(The authors gave the same response as above.)
